# Rapid Purification of Fucoxanthin from *Phaeodactylum tricornutum*

**DOI:** 10.3390/molecules27103189

**Published:** 2022-05-17

**Authors:** Xinjie Zhao, Liwei Gao, Xiangzhong Zhao

**Affiliations:** School of Food Sciences and Engineering, Qilu University of Technology (Shandong Academy of Sciences), Jinan 250353, China; zhaoxinjie1107@163.com (X.Z.); glw626816@163.com (L.G.)

**Keywords:** fucoxanthin, purification, SGCC, HLB solid-phase extraction, HPLC, ESI-MS, NMR, antioxidant

## Abstract

Fucoxanthin is a natural marine xanthophyll and exhibits a broad range of biological activities. In the present study, a simple and efficient two-step method was used to purify fucoxanthin from the diatom, *Phaeodactylum tricornutum*. The crude pigment extract of fucoxanthin was separated by silica gel column chromatography (SGCC). Then, the fucoxanthin-rich fraction was purified using a hydrophile–lipophile balance (HLB) solid-phase extraction column. The identification and quantification of fucoxanthin were determined by high-performance liquid chromatography (HPLC) and electrospray ionization mass spectrometry (ESI-MS). This two-step method can obtain 92.03% pure fucoxanthin and a 76.67% recovery rate. In addition, ^1^H and ^13^C NMR spectrums were adopted to confirm the identity of fucoxanthin. Finally, the purified fucoxanthin exhibited strong antioxidant properties in vitro with the effective concentration for 50% of maximal scavenging (EC50) of 1,1-Dihpenyl-2-picrylhydrazyl (DPPH) and 2,2′-Azinobis-(3-ethylbenzthiazoline-6-sulphonate) (ABTS) free radicals being 0.14 mg·mL^−1^ and 0.05 mg·mL^−1^, respectively.

## 1. Introduction

Fucoxanthin is a natural lutein which belongs to the oxygenated derivatives of carotenoids. Fucoxanthin possesses a unique structure, including an unusual allenic bond, a conjugated carbonyl group, 5,6-mono epoxide, and hydroxyl in the polyene chain of the molecule [1] (Figure 1). That exhibits many health beneficial properties due to its structural diversity, including antioxidant, anti-inflammation, anticancer, antiobesity, antidiabetic, antiangiogenic, antimalarial, hepatoprotective, cardioprotective, and neuroprotective activities [2,3,4]. Fucoxanthin is abundant in nature, accounting for about 10% of the total production of carotenoids [5]. However, the demand for fucoxanthin in the global market is rapidly growing and is expected to reach USD 120 million in 2022 due to the wide applications in food and medicine [6]. Moreover, fucoxanthin is considered safe by the European Food Safety Authority and the US Food and Drug Administration [7].

Fucoxanthin is mainly found in marine organisms, especially brown seaweeds, diatoms, macroalgae, and microalgae [8,9]. Marine microalgae are considered the best source of fucoxanthin because of their high photosynthetic efficiency, fast growth and reproduction, and high fucoxanthin content. *Phaeodactylum tricornutum* is a unicellular marine diatom rich in fucoxanthin and polyunsaturated fatty acids, such as eicosapentaenoic acid (EPA); fucoxanthin content is more than 10 times that of macroalgae [10]. The industrial production of fucoxanthin still faces challenges due to the complexity and low efficiency of chemical synthesis; therefore, fucoxanthin is mainly extracted and purified on a large scale from various brown algae [11]. At present, there are many methods for purifying fucoxanthin, and traditional methods include column chromatography, preparative thin-layer chromatography, and high-performance liquid chromatography [6,12]. With the development of modern chromatography technology, centrifugal partition chromatography and high-speed countercurrent chromatography have also been applied to the separation and purification of fucoxanthin [13,14]. Xia et al. [15] established a rapid extraction method for fucoxanthin in which it was separated and purified from crude pigment extractions of the marine diatom, *Odontella aurita*, by silica gel-based prepared high-performance liquid chromatography. The natural pigment extracts were eluted by open silica gel column chromatography with n-hexane: acetone (6:4, *v*/*v*) eluent, an orange-red-coloured fucoxanthin-rich fraction was separated, and the purity of fucoxanthin in the mixture was only 86.7%. Fucoxanthin with a purity of 97% was further obtained by preparative high-performance liquid chromatography. Raimundo et al. [16] used ultrasound-assisted ethanol extraction of fucoxanthin from *Tisochrysis lutea*. A two-step purification of fucoxanthin was optimized using centrifugal partition chromatography coupled to flash liquid chromatography. The purity of fucoxanthin that this process can obtain was as high as 99%. Ye et al. [17] adopted an eco-friendly method to purify fucoxanthin from the brown alga, *Sargassum horneri*. The fucoxanthin-rich ethanol extract was separated by octadecylsilyl column chromatography using ethanol: water (9:1, *v*/*v*) as the gradient eluent. Then, ethanol precipitation experiments were performed to obtain 91.07% purity fucoxanthin. Although that is an efficient method to purify fucoxanthin, it requires gradual precipitation during ethanol precipitation, which is time consuming and labour intensive. Numerous research during recent years [18,19] have also shown that a multi-step process is necessary to obtain high-purity fucoxanthin. In addition, most of them have been focused on high-end chromatographic techniques without sufficient attention to the development of efficient and miniaturized purification methods devoted to the sample treatment [20,21]. Therefore, it is necessary to develop a simple and rapid purification method.

The traditional SGCC purifies fucoxanthin, but it cannot achieve a perfect purification effect. Recently, solid-phase extraction has been widely used in the separation and purification of samples and may even be beneficial for unstable compounds [22]. More importantly, solid-phase extraction can significantly improve the accuracy and precision of analysis [23]. The simplest solid-phase extraction can be done manually: a syringe is connected to the upper end of the solid-phase extraction column. The sample in the queue is squeezed out of the column by pressing the needle. Alternatively, a positive or negative pressure solid-phase extraction unit can perform solid-phase extraction operations on bulk samples. With the development of technology and the increase in the number of pieces, more and more analytical laboratories have begun to use automatic solid-phase extraction instruments, especially multi-channel solid-phase extraction instruments used to process batch samples. The HLB solid-phase extraction column is widely used in the enrichment, separation, and purification of natural products and chemical drugs filled with polystyrene-divinylbenzene and allows for fast and simple sample preparation with reduced sample consumption and organic solvents [24].

Therefore, to avoid the shortcomings of traditional purification methods and modern chromatographic techniques, such as low-purity, high-equipment requirements and complicated processes, a simple and effective two-step method for the isolation and purification of fucoxanthin from *Phaeodactylum tricornutum* was established by combining traditional SGCC and modern solid-phase extraction techniques. After preliminary separation of the pigment crude extract of *Phaeodactylum tricornutum* by silica gel column, the obtained fucoxanthin-rich fraction was processed by HLB solid-phase extraction column in order to further purify and concentrate fucoxanthin. Quantification by HPLC and identification of purified compounds by ESI-MS and NMR spectroscopy demonstrated the high efficiency and feasibility of this method. In addition, the experiment of free-radical-scavenging ability confirmed that the purification method could not quench the antioxidant activity of fucoxanthin in vitro. This new purification method had the characteristics of rapidness, simplicity, low-equipment requirements, and high purity, and more importantly, it did not quench the original activity of purified components.

## 2. Results and Discussion

### 2.1. Purification and Identification of Fucoxanthin

The separation and purification procedure of fucoxanthin is illustrated in Figure 2. The crude pigment extracts from *Phaeodactylum tricornutum* were preliminarily eluted by SGCC, and different coloured pigment bands appeared on the silica gel column (Figure 2a). The principle for SGCC to separate different pigments is according to the various adsorption forces of components on silica gel. Pigments with greater polarity are easily adsorbed by silica gel and difficult to elute, while pigments with weaker polarity are not easily absorbed and eluted first [25]. The fucoxanthin content in various colour bands is shown in Figure 3a, and the orange-red part is rich in fucoxanthin. Therefore, the orange-red part was purified and enriched by the HLB solid-phase extraction column (Figure 2b). The HLB solid-phase extraction column was filled with polystyrene-divinylbenzene (Figure 2c) which contains a specific ratio of hydrophilic and hydrophobic groups: the hydrophobic divinylbenzene structure retains non-polar compounds, and the hydrophilic N-vinylpyrrolidone system keeps polar compounds; thus, it has balanced adsorption and good recovery for various polar and non-polar compounds [26].

A critical parameter of HLB solid-phase extraction column efficiency is the elution system. Five kinds of solvents with different polarities were compared to select the proper eluent, and the elution volume was also evaluated. As shown in Figure 3b, methanol provided the highest recoveries of fucoxanthin, followed by ethanol, acetone, and ethyl acetate, while n-hexane exhibited poor elution performance. Fucoxanthin is a type of lutein with great polarity, and the high-recovery rate of methanol elution was most likely related to its larger polarity. Additionally, methanol can be used directly for HPLC analysis since it is compatible with the mobile phase. Therefore, methanol was eventually chosen as the eluent. Among different elution volumes, the volume of 3 mL showed the highest recovery rate, and the solvents of 1 mL and 2 mL were insufficient to elute the target analytes. However, when the volume is increased to 5 mL, the recovery rate will decrease, which may be due to the dilution of the analyte leading to the decrease of sensitivity. Although 4 mL had the same recovery rate as 3 mL, 3 mL is chosen as the best elution volume for saving resources.

HPLC analyzed the pigment components of crude extracts and purified pigment, and the results are shown in Figure 4. The pigment eluting as a major peak at 11.09 min (Figure 4c) in the crude pigment extracts was identified as fucoxanthin according to the retention time of the fucoxanthin standard (Figure 4a) under the same conditions. The chromatogram showed that the crude pigment extracts were complex, and the fucoxanthin purity was 52.14% as calculated by the chromatography software. The chromatogram of purified fucoxanthin by SGCC and HLB solid-phase extraction column are shown in Figure 4b; under the optimal elution conditions, the purity of the fucoxanthin was identified as high as 92.03% with a 76.67% recovery rate. On the other hand, the purified compound’s ultraviolet-visible (UV-Vis) spectrum was recorded, and its absorption maximum (λ_max_) was compared with the fucoxanthin standard. From Figure 4d, the results showed that the standard and purified compounds exhibited the same spectroscopic profile with similar λ_max_. Except for the impurity absorption peak of fucoxanthin filtered by SGCC at 663 nm, all fucoxanthin showed characteristic absorption in the range of 350~550 nm, which was consistent with other research [27,28].

### 2.2. Characterization of Fucoxanthin by ESI-MS

Mass spectrometry can give a great deal of structural information on the bioactive compounds; therefore, we determined the molecular weight of purified fucoxanthin using ESI-MS. The exact molecular weight of fucoxanthin is calculated to be 658.4233 [M] according to the composition of its molecular elements. Results from Figure 5 indicate that the purified compound showed a characteristic peak at *m*/*z* 659.4301 [M+H]^+^. Furthermore, the spectrogram also showed the major ions at *m*/*z* 681.4112 [M+Na]^+^ and *m*/*z* 641.4186 [M+H-H_2_O]^+^ which were produced due to the combination of sodium ion and the loss of water from the characteristic peak ion (*m*/*z* 659.4301), respectively. These fragments were consistent with the mass fragmentation pattern reported by Xia et al. [15] and Raimundo Gonçalves et al. [16], wherein similar ions (*m*/*z* 659.8, 681.4131, 641.4222) were recorded for fucoxanthin. However, that was slightly different from the fragments data of fucoxanthin reported by de Quirós et al. [27] and Kim et al. [29]. Their research results showed the major fragments patterns of fucoxanthin at *m*/*z* 641.6 and *m*/*z* 581.6, respectively. The different results may be caused by different mass spectrometry conditions; ESI-MS ionization parameters have great effects on the component separation, fragmentation, and sensitivity [30]. The capillary voltage can influence the fragmentation pattern of sample ions, and the high-fragmentor voltage can also dissociate the compounds and increases the rate of fragmentation to generate a large number of smaller ions [30]. Gaurav Rajauria et al. [31] investigated the ionization behaviours and fragmentation pattern of fucoxanthin by changing the capillary voltage (2.0~4.5 kV) and fragmentor voltage (30~140 V). The results showed the best mass spectrum at the fragmentor voltage of 120 V, and the highest ion intensity appeared at the capillary voltage of 3.5 kV. Under this optimal ionization mode, the main fragments of fucoxanthin were produced at *m*/*z* 641.4 [M+H-H_2_O]^+^ and *m*/*z* 581.4 [M+H-H_2_O-AcOH]^+^.

### 2.3. NMR Analysis of Fucoxanthin

The conjugated double bonds are structurally very unstable; therefore, fucoxanthin can easily deteriorate by heat, oxygen, enzymes, and light during extraction and purification [32,33]. The instability of fucoxanthin can lead to structural isomerization and even oxidative cleavage of the backbone, which may further lead to the loss of its biological activity [11]. Additionally, *Phaeodactylum tricornutum* contains many carotenoids with similar structures, and their existence will affect the purification, quantification, and identification of fucoxanthin. Therefore, to confirm the molecular structure of purified compounds, NMR spectroscopy was utilized.

The results of the ^1^H and ^13^C NMR spectra indicated that the purified compound was a polyene structure containing two quaternary geminal dimethyl, four olefinic methyls, conjugated ketone, and acetyl functional groups. The chemical structures above indicated that the purified compound was a carotenoid. The complete structural elucidation data of ^1^H and ^13^C NMR spectra are listed in Table 1. These data are consistent with previously published results [31,34]. The ^1^H and ^13^C NMR spectrum of the purified compounds (Figure 6) were identical with those of the authentic fucoxanthin standard, which confirmed their identity as fucoxanthin.

### 2.4. Antioxidant Activity

To verify the biological activity of fucoxanthin purified from *Phaeodactylum tricornutum*, the antioxidant capacity experiment of fucoxanthin was carried out using DPPH and ABTS free radicals. The fucoxanthin standard was used as a reference standard to compare the results and tocopherol as a positive control. The free-radical-scavenging effect was expressed as “concentration for 50% of maximal effect (EC_50_)”, which is the sample concentration required to scavenge 50% of free radicals. Results show that the purified fucoxanthin exhibited similar antioxidant capacity against DPPH and ABTS free radicals compared to the fucoxanthin standard. The scavenging ability of fucoxanthin for DPPH free radicals gradually increased as the concentration of fucoxanthin increased from 0.02 mg·mL^−1^ to 0.2 mg·mL^−1^ (Figure 7a). The scavenging ability of the fucoxanthin standard for DPPH free radicals was almost the same as tocopherol at 0.2 mg·mL^−1^, but for purified fucoxanthin, the scavenging ability for DPPH free radicals was only 82.23% of tocopherol, which may be related to the purity of purified fucoxanthin. The impurities present in purified fucoxanthin limited the expression of fucoxanthin activity. Both the fucoxanthin standard and purified fucoxanthin scavenged the DPPH free radicals in a dose-dependent manner with EC_50_ values of 0.13 mg·mL^−1^ and 0.14 mg·mL^−1^, respectively. The scavenging effect of purified fucoxanthin for ABTS free radicals was higher than that of DPPH free radicals (Figure 7b). In the concentration range of 0.12 mg·mL^−1^ to 0.2 mg·mL^−1^, the fucoxanthin standard and tocopherol had the same scavenging capacity for ABTS free radicals, while the purified fucoxanthin reached 95.66% of tocopherol at 0.2 mg·mL^−1^. The ABTS-scavenging activity was linearly dependent on the fucoxanthin concentration, and the EC_50_ for ABTS free radicals was 0.05 mg·mL^−1^. The EC_50_ was 0.03 mg·mL^−1^ for the fucoxanthin standard.

It has been confirmed that fucoxanthin can effectively quench chemically generated DPPH free radicals and show strong scavenging ability against them; however, the exact mechanism of action is still unclear [35,36]. Previous studies have established that fucoxanthin exhibits antioxidant properties based on its ability to quench singlet oxygen and trap free radicals. Unlike other carotenoids, such as β-carotene and lutein, fucoxanthin is an effective singlet oxygen quencher [4,37]. DPPH free radicals have single electrons. Fucoxanthin reduces DPPH free radicals by donating electrons to pair single electrons, while other antioxidants usually act as proton donors [38]. Second, the presence of multiple conjugated double bonds in the fucoxanthin structure facilitates its quenching of singlet oxygen and free radical scavenging; it was speculated that fucoxanthin transfers excited electrons to conjugated double bonds to generate more stable free radicals and excited states of carotenoids, and in addition to conjugated double bonds, allene bonds and hydroxyl groups in fucoxanthin equally have the effect of scavenging free radicals [36]. Furthermore, it has been reported that the anticancer effects of fucoxanthin are closely related to its antioxidant activity or pro-oxidative development [9]. Reactive oxygen species (ROS), mainly including hydroxyl radicals, superoxide radicals, and hydrogen peroxide, can cause oxidative stress to intracellular lipids, proteins, and DNA, induce oxidative stress, and lead to various tumours and diabetes, liver damage, and central nervous system diseases. Antioxidants can scavenge multiple reactive oxygen species produced in the body to prevent the generation of reactive oxygen species-induced oxidative stress [1,38,39].

## 3. Materials and Methods

### 3.1. Chemicals and Reagents

Chromatography-grade methanol was obtained from Macklin Biochemical (Shanghai, China). The analytical reagents, methanol, ethanol, n-hexane, and ethyl acetate, were purchased from Sinopharm (Beijing, China). Ultrapure water was produced by a Millipore Milli-Q system (Billerica, MA, USA). 

### 3.2. Cultivation and Collection of Microalgae

The algae species of *Phaeodactylum tricornutum* (GY-H9) was purchased from Guangyu Biological Technology Co. (Shanghai, China). The inoculum of algae cells was cultivated in artificial seawater fortified with F/2+Si medium at 20 °C ± 1 in an artificial climate incubator according to the description of Hao [40]. The pH of the diatom medium was adjusted to 8.0 before autoclaving at 121 °C for 20 min. After 14 days of cultivation under a light cycle of 12 h light and 12 h dark, the cultured microalgae were collected by centrifugation at 4000 rpm for 10 min and then lyophilized. Lyophilized powders were stored at −20 °C for later analysis.

### 3.3. Extraction of Total Pigments from Phaeodactylum Tricornutum

The total pigments were determined according to the method of Sachindra [36] with minor modification. *Phaeodactylum tricornutum* lyophilized powders (1 g) were subjected to sonication-assisted extraction using absolute ethanol (30 mL) as a solvent for 60 min. Ethanol extracts were centrifuged for 15 min at 6000 rpm at 4 °C in the centrifuge (Hitachi, Tokyo, Japan), and the supernatants were collected. Precipitation was re-extracted with ethanol (30 mL) until their supernatants became colourless. The extraction procedure was entirely performed at 45 °C. The supernatants were then combined and vacuum concentrated at 35 °C by a rotary evaporator (Yarong, Shanghai, China), and the ethanol extracts were stored at −70 °C for the following assays. All procedures were carried out under darkness to avoid photo-oxidation.

### 3.4. Isolate the Crude Pigment Extracts of Fucoxanthin by SGCC

The ethanol extracts were isolated through the silica gel column chromatography according to previous report [41] with some modifications. The column chromatography silica gel (50 g, 200~300 mesh, Qingdao Haiyang Chemical, Qingdao, China) was mixed with n-hexane (100 mL) and packed into a glass column (2.5 × 40 cm), and then eluted with a mixture of n-hexane: ethyl acetate: methanol (7:2:1, *v*/*v*). The fucoxanthin-rich fraction was collected and concentrated into a small volume for the next purification.

### 3.5. Purification of Fucoxanthin by HLB Solid-Phase Extraction Column

The fucoxanthin-rich concentrate was purified by the HLB solid-phase extraction column (60 mg/3 mL, Merck, Shanghai, China). In this purification procedure, the application method of the solid-phase extraction column was as follows according to the description of Qi [26]: the HLB solid-phase extraction column was first activated with 3 mL methanol and then equilibrated with 3 mL ultrapure water. The sample solution was then gravity-loaded onto the cartridge, and the effluent was discarded. After rinsing the cartridge with 3 mL ultrapure water to remove impurities, the retained target analytes were eluted with a certain amount of organic solvent. Finally, the eluate was ready for the following analysis.

### 3.6. HPLC Analysis of Fucoxanthin

Fucoxanthin analysis was performed on a Shimadzu LC-20A system (Kyoto, Japan) equipped with a variable-wavelength UV-visible light detector (SPD-20A), a system controller (CBM-20A), an auto-sampler (SIL-20A), a column oven (CTO-20A), tandem double plunger (LC-20AT), and LabSolutions software. An Inert Sustain C18 reverse-phase column (5 μm particle size, 4.6 × 250 mm ID, GL Sciences, Tokyo, Japan) was used.

The HPLC conditions were set as follows according to the description of Liu [41] with a minor modification: the mobile phases of methanol: water (89:11, *v*/*v*) and the elution program were performed using the isocratic mode. The elution was performed at a flow rate of 1.0 mL·min^−1^ for 20 min with 20 μL injection volume and 35 °C column temperature.

All chromatographic data were recorded from the 190 to 700 nm range, and the determination of fucoxanthin was carried out according to the absorbance at 450 nm absorption wavelength. A regression equation (Table 2) was established for the quantification of fucoxanthin by HPLC using the fucoxanthin standard (purity ≥ 98%, Sigma-Aldrich, Shanghai, China). The calibration curves calculated the amounts of fucoxanthin from the peak areas. The fucoxanthin concentrations in the microalgal samples were expressed as milligram fucoxanthin per gram cell dry weight (mg·g^−1^ DW).

### 3.7. ESI-MS Analysis of Fucoxanthin

ESI-MS analysis was performed with a Thermo Fisher Scientific Q Exactive Focus Combined Quadrupole Orbitrap Mass spectrometer (Waltham, MA, USA) fitted with MassHunter Workstation software. The MS conditions were set as follows according to the description of Raimundo Gonçalves [16]: positive ions mode, capillary voltage 3.0 kV, sampling cone voltage 35 V, source compensation 80 V, and collision energy 10 eV. The nebulizing and drying gas used was nitrogen at a pressure of 50 psi, flow rate of 5 L min^−1^, and drying temperature of 500 °C. Mass spectral data were recorded in the mass range of *m*/*z* 60~900. The sampling method adopted direct injection and mass spectrometry was calibrated before analysis using 0.5 mmol sodium formate solution.

### 3.8. Identification of Fucoxanthin by NMR 

The purified fucoxanthin was evaporated under a vacuum at 45 °C to remove methanol and was dried under a nitrogen stream to remove residual methanol solvent. The sample (10 mg) was redissolved in 1.2 mL of deuterated (d) chloroform, and then this sample was transferred into a 5 mm NMR sample tube and analyzed. The ^1^H NMR spectra and ^13^C NMR spectra were performed on purified compound at 400 MHz and 100 MHz frequency, respectively, using the Bruker 400 MHz NMR system (Karlsruhe, Germany); the spectra were scanned for 1 h and 6 h, respectively, and at room temperature. Chemical shifts (δ) represent the shifts of the resonance absorption peaks in ppm relative to the chemical shifts of the solvent signals, whereas the coupling constants (J) are reported in Hertz (Hz). Data were processed using the MestReNova11 program (Mestrelab Research, Santiago de Compostela, Spain).

### 3.9. Assay for Antioxidant Activity of Fucoxanthin

Antioxidants are substances that trap, neutralize, and remove free radicals, thereby preventing the oxidation of macromolecules [42]. DPPH and ABTS free radicals are stable radical sources for evaluating the free-radical-scavenging ability of antioxidant components in vitro scavenging ability of free radicals [43]. The scavenging activity of DPPH of fucoxanthin was determined according to the method of Sachindra [36] with a minor modification. Briefly, 0.5 mL fucoxanthin solution (0.02~0.2 mg·mL^−1^) was mixed with an equal volume of 0.50 mmol·L^−1^ DPPH solution in ethanol. The mixture was shaken quickly and reacted for 30 min at room temperature in the dark after mixing. The absorbance was measured at 517 nm. The scavenging capacity was calculated according to the following equation:
DPPH free-radical-scavenging capacity (%) = [1 − (A_1_ − A_2_)/A_0_] × 100
where A_0_ is the absorbance when using absolute ethanol instead of fucoxanthin, A_1_ is the absorbance of different concentrations of fucoxanthin solution reacted with DPPH free radicals, and A_2_ is the absorbance after using absolute ethanol instead of DPPH to react with different concentrations of fucoxanthin solutions.

The scavenging activity of ABTS free radicals was measured, as described by Osman [44], with minor modifications. The ABTS free radicals stock solution was generated by mixing 7.40 mmol·L^−1^ ABTS diammonium salt and 2.60 mmol·L^−1^ potassium persulphate in a ratio of 1:1 and reacting at room temperature for 12 h in the dark. The stock solution was diluted with absolute ethanol until the absorbance at 734 nm reached 0.7 ± 0.05 units. Then, the ethanolic fucoxanthin solution (0.02~0.2 mg·mL^−1^) and the diluted ABTS free radicals working solution were mixed in a ratio of 1:3; the mixture was incubated at 30 °C for 15 min in a water bath. The absorbance at 734 nm was measured. The scavenging capacity was calculated according to the following equation:
ABTS free-radical-scavenging activity (%) = [(A_0_ − A)/A_0_] × 100
to obtain concentrations of fucoxanthin where A_0_ is the absorbance when using absolute ethanol instead of fucoxanthin, and A is the absorbance of different solution reacted with ABTS free radicals.

### 3.10. Statistical Analysis

Data were expressed as mean ± standard error of the mean and analyzed using IBM SPSS Statistics 21. All charts were processed by Origin 2017 and MS Office.

## 4. Conclusions

This present study explored a strategy for the rapid isolation of fucoxanthin from the diatom, *Phaeodactylum tricornutum*. Fucoxanthin was extracted from a mixture of ethanol extracts, and the crude pigment extracts were purified by SGCC and HLB solid-phase extraction column. The purified fucoxanthin was identified and characterized by HPLC and ESI-MS. In addition, NMR spectroscopy and the data was compared with an authentic fucoxanthin standard. In this study, a combination of traditional SGCC and HLB solid-phase extraction columns was used to purify carotenoids with good purity, which provided new insights into the purification method of fucoxanthin. The free-radical-scavenging experiments showed that the identified fucoxanthin had strong antioxidant activity. The results of this study may contribute to the rapid separation and purification of fucoxanthin.

## Figures and Tables

**Figure 1 molecules-27-03189-f001:**
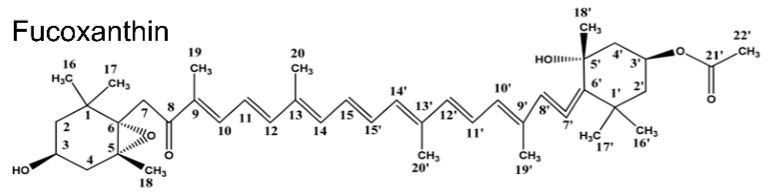
Chemical structure of fucoxanthin.

**Figure 2 molecules-27-03189-f002:**
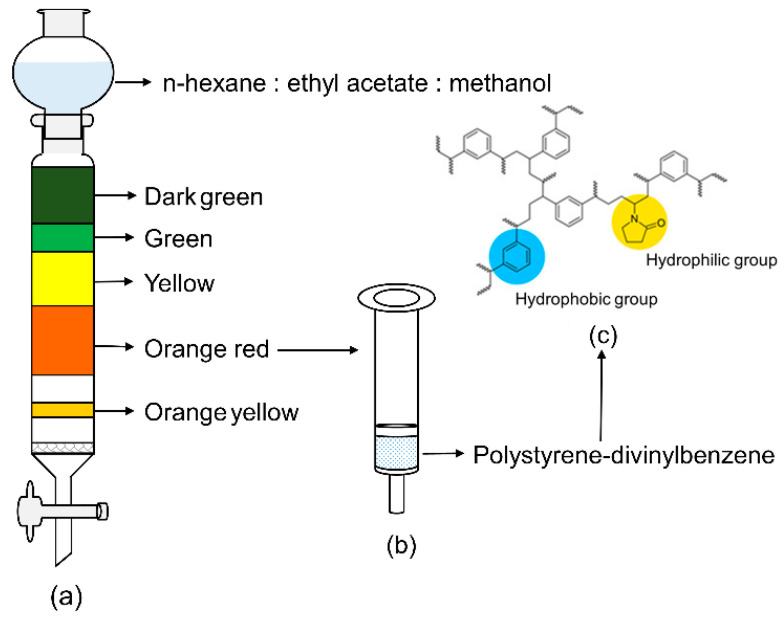
Isolation and purification of fucoxanthin from crude pigments extraction of *Phaeodactylum tricornutum*. (**a**) SGCC; (**b**) HLB solid-phase extraction column; (**c**) chemical structure of polystyrene-divinylbenzene.

**Figure 3 molecules-27-03189-f003:**
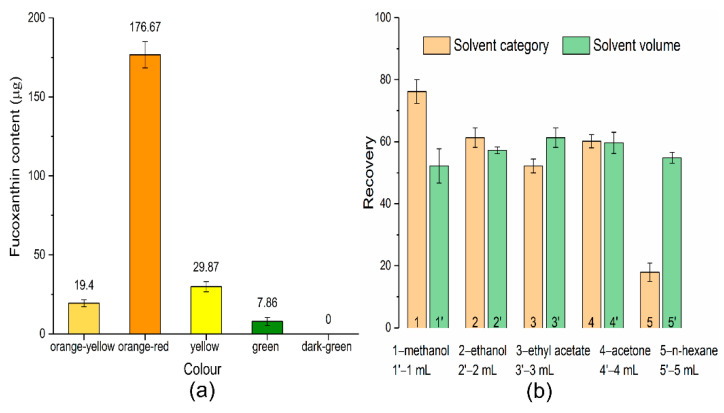
Fucoxanthin content in different colour bands (**a**), elution solvent and volume on the recovery rate of HLB solid-phase extraction column (**b**).

**Figure 4 molecules-27-03189-f004:**
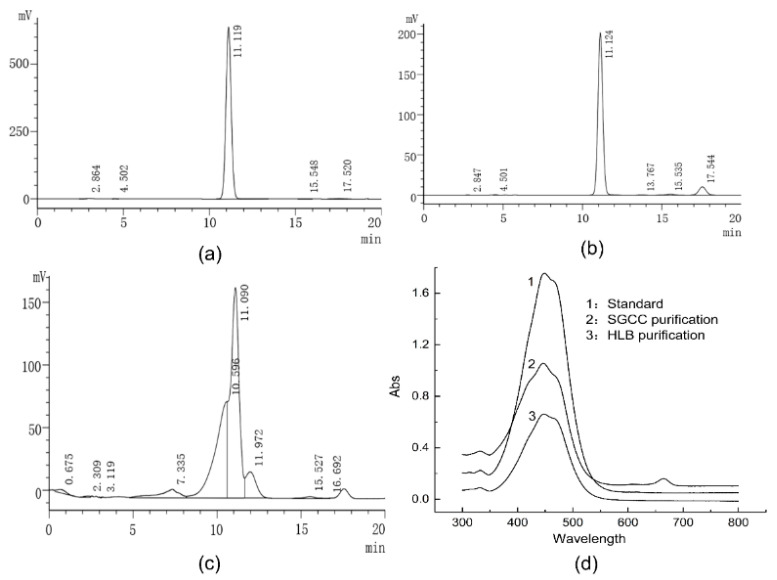
HPLC and UV-Vis spectra of fucoxanthin: (**a**) liquid chromatogram of fucoxanthin standard; (**b**) liquid chromatogram of purified fucoxanthin; (**c**) liquid chromatogram of the crude extracts of fucoxanthin; (**d**) UV-Vis spectrogram of fucoxanthin standard (1), SGCC purified fucoxanthin (2), and HLB purified fucoxanthin (3).

**Figure 5 molecules-27-03189-f005:**
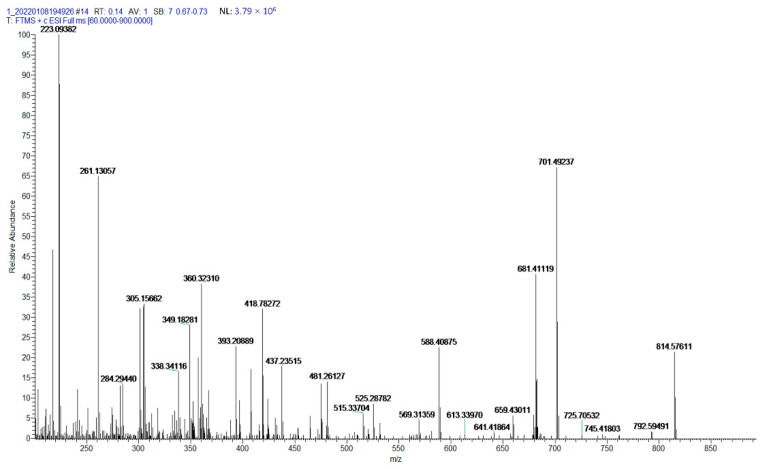
ESI-MS spectra in full-scan mode from *m*/*z* 60 to 900 of purified fucoxanthin.

**Figure 6 molecules-27-03189-f006:**
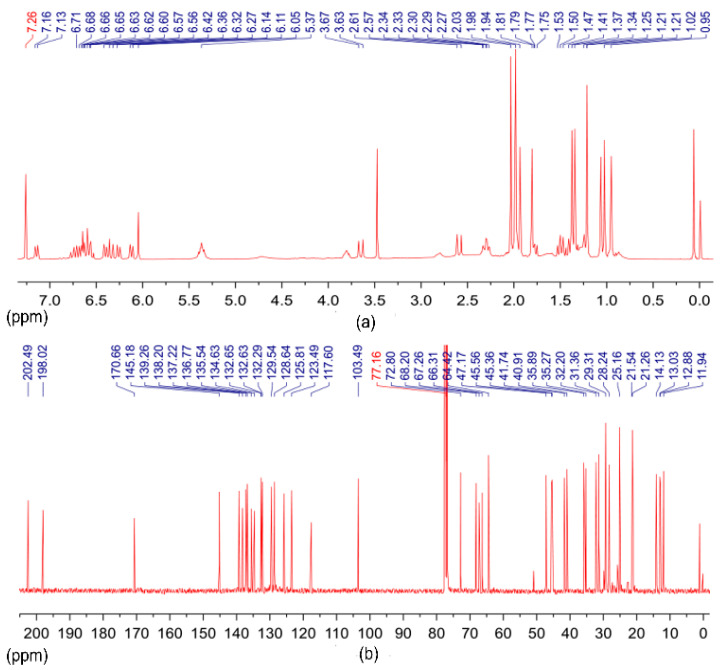
NMR spectra of the purified compounds: (**a**) ^1^H NMR spectrum; (**b**) ^13^C NMR spectrum.

**Figure 7 molecules-27-03189-f007:**
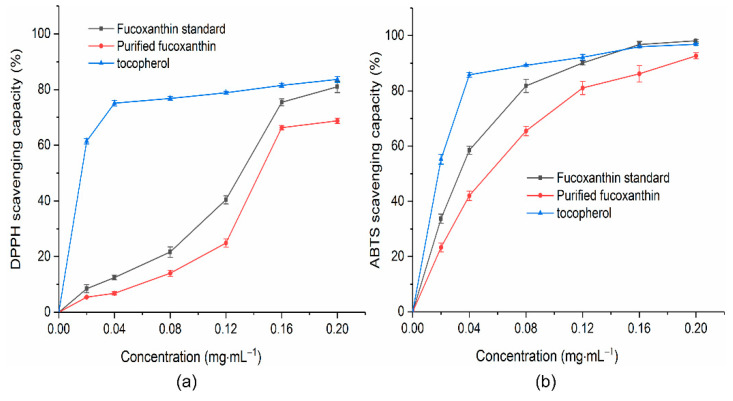
Antioxidant assays for the purified fucoxanthin from *Phaeodactylum tricornutum*: (**a**) scavenging capacity of DPPH free radicals; (**b**) scavenging capacity of ABTS free radicals. EC_50_ values were calculated by nonlinear regression analysis.

**Table 1 molecules-27-03189-t001:** ^1^H and ^13^C NMR spectral data of purified fucoxanthin recorded in CDCl_3_ at 400 MHz and 100 MHz respectively.

Position	δ_C_, Type	δ_H_ (*J* in Hz)	Position	δ_C_, Type	δ_H_ (*J* in Hz)
1	35.89, C		1′	35.27, C	
2	47.17, CH_2_	1.42, *m* 1.61, *m*	2′	45.56, CH_2_	1.49, *m* 2.07, *m*
3	64.42, CH	3.80, *m*	3′	68.20, CH	5.37, *m*
4	41.74, CH_2_	1.77, *m* 2.29, *m*	4′	45.36, CH_2_	1.67, *m* 2.33, *m*
5	66.31, C		5′	72.80, C	
6	67.26, C		6′	117.60, C	
7	40.91, CH_2_	2.59, *d* (18.3) 3.65, *d* (18.4)	7′	202.49, C	
8	198.02, C=O		8′	103.49, CH	6.05, *s*
9	134.63, C		9′	132.63, C	
10	139.26, CH	7.14, *d* (10.8)	10′	128.64, CH	6.12, *d* (11.3)
11	123.49, CH	6.64, *m*	11′	125.81, CH	6.74, *d* (11.5)
12	145.18, CH	6.70, *d* (11.1)	12′	137.22, CH	6.34, *d* (15.0)
13	138.20, C		13′	135.54, C	
14	136.77, CH	6.40, *d* (11.5)	14′	132.29, CH	6.26, *d* (11.5)
15	129.54, CH	6.56, *dd* (14.8, 11.0)	15′	132.65, CH	6.78, *m*
16	25.16, CH_3_	1.02, *s*	16′	29.31, CH_3_	1.34, *s*
17	28.24, CH_3_	0.95, *s*	17′	31.36, CH_3_	1.25, *s*
18	21.26, CH_3_	1.21, *s*	18′	32.20, CH_3_	1.37, *s*
19	11.94, CH_3_	1.94, *s*	19′	14.13, CH_3_	1.81, *s*
20	12.88, CH_3_	1.98, *s*	20′	13.03, CH_3_	1.98, *s*
			21′	170.66, C=O	
			22′	21.54, CH_3_	2.03, *s*

**Table 2 molecules-27-03189-t002:** The relationship between the fucoxanthin concentration and the peak area.

Regression Equation	Correlation Coefficient	Linearity Range
Y = −388,741.28 + 215,585.34X	R^2^ = 0.9994	10~200 μg·mL^−1^

## Data Availability

Not applicable.

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
