# Peer review of "Rapid Purification of Fucoxanthin from *Phaeodactylum tricornutum"

_molecules, 2022, doi:10.3390/molecules27103189_

Round 1

Reviewer 1 Report

The manuscript presents a simple and efficient method to purify fucoxanthin. However, the authors should consider reinforcing the novelty in the introduction section.

 Also, the discussion section is poor. I recommend that the authors should reinforce this section. 

The keywords should be modified because they are in the title. 

The authors should clarify the main goal in the document.

Finally, the authors should include references in the methodologies.  

Author Response

Point 1: the authors should consider reinforcing the novelty in the introduction section. Also, the discussion section is poor. I recommend that the authors should reinforce this section.  The keywords should be modified because they are in the title. The authors should clarify the main goal in the document. Finally, the authors should include references in the methodologies. 

Response 1: We have strengthened the novelty of the article and clarified the main objectives of the study in the introduction. The key words have indeed been included in the title of the article, so they have also been revised. The discussion was also strengthened according to the experimental content. Finally, the references of methods have been supplemented in sections 3.2, 3.3, 3.4, 3.5 and 3.6, however, the method of NMR in section 3.8 was determined according to the specific operating conditions during the experiment.

Reviewer 2 Report

The present paper described an efficient method to purify fucoxanthin, and its identification and quantification.  The analytical methods, such as SGCC, HPLC, HPLC, ESI-MS and 1H NMR spectroscopy, were used. 

The following points should be considered for publication:

  • Line 16: “1H nuclear” should be “1H nuclear”.

  • Line 18: “in vitro” should be italic.

  • Line 156: “1H NMR spectra signal” should be “1H NMR spectra”.

  • Line 160: Regarding 1H assignments and structure determination of fucoxanthin, a comparison of 1H chemicalshifts between the present compound and the authentic compound was only applied. Both NMR spectra should be shown for clarity.  Were other conventional 2D NMR methods employed for the complete assignments?  The simple comparison of 1H NMR spectra cannot be a strong evidence for the structure determination.

  • Line 163: In Table 1, notations such as “s, m” should be italic.

  • Line 278: Regarding solvent signals “, what was the NMR solvent? The measuring condition of NMR, such as temperature and concentration, must be described.

  • The list of abbreviations, such as DPPH, ABTS, HLB and SGCC, should be described for convenience.

Author Response

Point 1: Line 16: “1H nuclear” should be “1H nuclear”. 

Line 18: “in vitro” should be italic. 

Line 202: “1H NMR spectra signal” should be “1H NMR spectra”. 

Line 210: In Table 1, notations such as “s, m” should be italic.

Response 1: The above questions you raised have been revised and marked in the manuscript.

Point 2: Line 206: Regarding 1H assignments and structure determination of fucoxanthin, a comparison of 1H chemicalshifts between the present compound and the authentic compound was only applied. Both NMR spectra should be shown for clarity.  Were other conventional 2D NMR methods employed for the complete assignments?  The simple comparison of 1H NMR spectra cannot be a strong evidence for the structure determination

Response 2:

According to your opinion, the NMR spectrum of the purified compound was supplemented in Section 2.3 of the manuscript, however the NMR spectrum of the authentic compound was based on the test report provided by the purchased company, so it can't be well presented in the manuscript, but it has been uploaded to the attachment for your review.

The simple comparison of 1H NMR spectra can not be a strong evidence for the structure determination, so the 13C NMR spectra was supplemented to confirm the structure of purified compound. 2D NMR plays an important role in inferring the structure of complex compounds which is difficult for ordinary NMR to analyze. Although fucoxanthin is a high molecular weight pigment with various functional groups, ordinary NMR is enough to analyze its structure. Therefore, 2D NMR has not been used to characterize the structure of fucoxanthin in this study.

Point 3: Line 350: Regarding solvent signals “, what was the NMR solvent? The measuring condition of NMR, such as temperature and concentration, must be described.

Response 3: The “solvent signals” refers to the NMR signals of deuterated solvents. In this study, deuterated chloroform was used as the solvent for NMR test samples. The 1H and 13C spectrum signals of the tested samples were expressed as "chemical shift", and the chemical shift is a relative value, which is expressed as the frequency difference between the signal of the tested sample and the signal of the standard substance (tetramethylsilane, TMS, δTMS=0) or deuterated solvent. The deuterated solvents can lock the field accurately and avoid the interference of hydrogen atoms in common solvents.

The measuring conditions of NMR have been supplemented in section 3.8 of the manuscript.

Point 4: The list of abbreviations, such as DPPH, ABTS, HLB and SGCC, should be described for convenience.

Response 4: The list of abbreviations is a clear expression way for all the abbreviations in the manuscript, and it is a good choice. However, according to the requirements of “Molecules” and “ACS Publishing Center”: “Authors should not provide a separate list of abbreviations in a manuscript.”, therefore, we didn't add a list of abbreviations in the manuscript.

Reviewer 3 Report

The manuscript entitled Rapid purification of fucoxanthin from Phaeodactylum tricornutum by Xinjie Zhao, Liwei Gao and Xiangzhong Zhao deals with the isolation of fucoxantin which is simple, well designed and rapid, which is positive. On the other hand, antioxidative activity tests used are old known methods (DPPH and ABTS tests) and not supported with some newer antioxidative method (in vivo). Some antioxidative lipophilic standard like tocopherol for example should be used, not only comparison with the fucoxantin standard. My opinion is that this manuscript is not suitable for this journal and I recommend rejection. Some analytical chemistry or food chemistry journals should be considered.

Author Response

Point 1: antioxidative activity tests used are old known methods (DPPH and ABTS tests) and not supported with some newer antioxidative method (in vivo).

Response 1: We understand that antioxidant experiments in vivo may better demonstrate the antioxidant activity of fucoxanthin. However, due to the time limit, we can't complete the antioxidant experiment in vivo of fucoxanthin in a short time. In the current research, the main purpose of antioxidant experiment using DPPH and ABTS was to confirm the influence of the new methods on the functional activity of the purified fucoxanthin. Although these old testing methods are not the best choice, they can also prove the effectiveness of the purification methods to some extent.

Point 2: Some antioxidative lipophilic standard like tocopherol for example should be used, not only comparison with the fucoxantin standard.

Response 2: According to your suggestion, we used tocopherol to carry out antioxidant experiment,  and used it as a positive control to evaluate the purified fucoxanthin. The specific analysis has been described in section 2.4 of the manuscript.

Point 3: My opinion is that this manuscript is not suitable for this journal and I recommend rejection. Some analytical chemistry or food chemistry journals should be considered.

Response 3: The main content of this manuscript is to establish a simple method to purify fucoxanthin from seaweed, and to identify the purified fucoxanthin from the molecular structure level. This process involves the field of analytical chemistry. As you said, this research is suitable for journals of analytical chemistry and food chemistry. However, Molecules is an open access journal of synthetic organic chemistry and natural product chemistry, which main research areas include (but are not limited to): Supramolecular chemistry, Medicinal chemistry, Analytical chemistry, Inorganic chemistry and Natural products. so I think this manuscript is suitable for Molecules.

Round 2

Reviewer 2 Report

The reply to the comments has been reviewed, and it can be considered

 that sufficient revisions have been made in the revised manuscript.

Reviewer 3 Report

The revised manuscript “Rapid purification of fucoxanthin from Phaeodactylum tricornutum” by Xinjie Zhao, Liwei Gao and Xiangzhong Zhao is significantly improved original manuscript about isolation of large amounts of fucoxantin which is simple, but well designed and rapid. I recommend acceptance after minor revision.

Corrections:

Abstract

Line 16: Put “1H and 13C NMR spectra” instead of “1H and 13C nuclear magnetic resonance (NMR) spectrums”

Introduction

Line 88: remove “…in this study.”

Line 90: Put “…fucoxantin rich fraction…” instead of “…fucoxantin rich part…”

Line 94: Put “…in order to further purify and concentrate fucoxantin.” instead of “… which can further purify fucoxantin and enrich fucoxantin simultaneously.”

Line 94: Put “Quantification by HPLC and identification of purified compounds by ESI-MS and NMR…”

Lines 97,100: Use “quench” instead of “destroy”

Results and Discussion

Line 126: Put “Fucoxantin is a type of…” instead of “Fucoxantin is a kind of…”

Line 139: Put “11.09 min” instead of “11.090 minutes”

Line 169: Put “major ions” instead of “major fragments patterns”; [M+Na]+ ion is not a fragment

Line 171: Put “respectively” at the end of the sentence.

Line 173: Remove “fragment”

Line 175: Put ”…Kim et al. [30]. Their…”    instead of “…Kim et al. [30], their…”

Line 208: Add “standard” after “authentic fucoxantin”

Line 211: Put “in CDCl3” after “recorded”

Line 217: Put “NMR spectra…” instead of “The 1H and 13C NMR spectrum…” at the beginning of the Fig 6. Caption

Line 232. Put “present” instead of “contained”

Line 277: change font

Materials and Methods

Lines 344 and 345. Remove “proton”, brackets and “carbon”

Author Response

The above questions provided by reviewer 3 in the second round of review have been revised and marked in the manuscript.